# Dynamic Intercell Communication between Glioblastoma and Microenvironment through Extracellular Vesicles

**DOI:** 10.3390/biomedicines10010151

**Published:** 2022-01-11

**Authors:** Gloria Krapež, Katarina Kouter, Ivana Jovčevska, Alja Videtič Paska

**Affiliations:** Institute of Biochemistry and Molecular Genetics, Faculty of Medicine, University of Ljubljana, SI-1000 Ljubljana, Slovenia; gloria.krapez@mf.uni-lj.si (G.K.); katarina.kouter@mf.uni-lj.si (K.K.)

**Keywords:** microenvironment, inflammation, exosome, therapy

## Abstract

Glioblastoma is simultaneously the most common and most aggressive primary brain tumor in the central nervous system, with poor patient survival and scarce treatment options. Most primary glioblastomas reoccur and evolve radio- and chemoresistant properties which make them resistant to further treatments. Based on gene mutations and expression profiles, glioblastoma is relatively well classified; however, research shows that there is more to glioblastoma biology than that defined solely by its genetic component. Specifically, the overall malignancy of the tumor is also influenced by the dynamic communication to its immediate and distant environment, as important messengers to neighboring cells in the tumor microenvironment extracellular vesicles (EVs) have been identified. EVs and their cargo can modulate the immune microenvironment and other physiological processes, and can interact with the host immune system. They are involved in tumor cell survival and metabolism, tumor initiation, progression, and therapy resistance. However, on the other hand EVs are thought to become an effective treatment alternative, since they can cross the blood–brain barrier, are able of specific cell-targeting and can be loaded with various therapeutic molecules.

## 1. The Genetic Component of Glioblastoma

Glioblastoma is simultaneously the most common and most aggressive primary brain tumor in the central nervous system [1]. It accounts for as much as 80% of all malignant primary brain tumors and more than 60% of all adult brain tumors [2]. Despite countless strides in the field of medicine and brain malignancies, the patient survival continues to stay dismal, averaging at around 15 months after diagnosis [2,3,4]. Only about 4–6.7% of patients see the 5-year check mark [3,4].

Even though glioblastoma has been recognized and researched for the past 150 years, since it was histomorphologically described by Rudolf Virchow, the comprehension and successful treatment options remain scarce to this day [5,6]. Current standard of care for the majority of newly diagnosed patients involves surgery and maximal safe tumor resection followed by field radiotherapy and concomitant chemotherapy with temozolomide (TMZ) [7]. TMZ is an alkylating agent that uses common CpG methylations of the promotor region of the O^6^-methylguanin DNA methyltransferase (MGMT) in glioblastoma cells, to institute cell-cycle arrest which leads to cell apoptosis [8]. So far this is the only proven therapeutic intervention that prolongs overall survival of patients with newly diagnosed glioblastoma. Unfortunately, most primary glioblastomas reoccur and evolve radio- and chemoresistant properties which make them resistant to further treatments [7].

There are a lot of reasons for the dismal prognosis of patients. Some of them include therapy resistant glioblastoma stem cells that can repopulate destroyed glioblastoma cells after radio- and chemotherapy. Glioblastoma stem cells can also produce vascular endothelial growth factor (VEGF) that promotes angiogenesis and thus create a favorable microenvironment for their own survival [9]. High intra- and inter-tumoral heterogeneity provides additional challenges when implementing new or existing therapies, since they usually cause resistances in surviving glioblastoma cells [10]. Genetic changes of glioblastoma cells often initiate higher metabolic and proliferative capability, which in turn lead to high invasiveness deeper into its surrounding parenchyma [10,11]. Dispersed glioblastoma cells in the parenchyma are usually impossible to fully surgically excise and result in reoccurrences after treatment [12].

In addition to previously mentioned reasons for low life expectancy, the fact that glioblastoma mainly impacts elderly patients that have lower tolerance for current treatment options leads to poorer outcomes. Advanced age is regarded to be one of the most important negative prognostic factors when assessing life expectancy of patients [13]. A study in the United States showed that the average incidence rises with age. If we consider the whole population of the United States, the incidence stands at around 3.2 per 100,000 people [4,13], which is in accord with the worldwide average incidence that ranges from 0.59 to 5 per 100,000 people [14]. If we age adjust occurrences of glioblastoma, we see that the incidence rate rises to 15.47 per 100,000 people in age group ranging from 75 to 79 years old [13]. The same thing was pointed out when Walker et al. looked at the Canadian population between the years 2009 and 2013. The average incidence rate for people older than 65 years was 13.34 in comparison to other age groups where the incidence rate never rose above 8.9 [15]. Because of this and the increase in longevity among the developed countries, the median age of people diagnosed with glioblastoma is around 64 years [4,13]. Glioblastoma is also more prevalent among males than females in all age groups with an average ratio of 1.6:1 [16].

Glioblastoma falls under the umbrella term “adult-type diffuse glioma” that also encompass astrocytoma and oligodendroglioma [17]. In the older 2016 World Health Organization (WHO) classification of gliomas, glioblastoma was divided based on the isocitrate dehydrogenase (*IDH*) mutation into *IDH* mutated or *IDH* wild type [18]. This prognostic marker still stands as a very important characteristic when predicting patient survival, since mutations in *IDH 1* or *2* often lead to a better prognosis in patients [19]. The WHO released a revised glioma classification in 2021 that reclassified glioblastoma. Tumors that present with an *IDH* mutation are now classified in the astrocytoma category [17,20]. The definition of glioblastoma now includes all astrocytic gliomas that do not have mutated *IDH* and *H3* genes and also include one or more characteristically genetically altered profile involving TERT promoter, *EGFR* gene amplification, +7/−10 chromosome copy-number alterations or present themselves with necrosis or microvascular proliferation [20]. Glioblastoma is still characterized as a grade IV tumor on the WHO central nervous system tumor grading scale, which represent the most invasive malignancies with the worst prognosis [17].

Based on gene expression profiles, glioblastoma is further divided into 4 subtypes: classical, mesenchymal, neural, and proneural. This classification was proposed by Verhaak et al. in 2010 [21]. In 2017 Wang et al. challenged this subdivision of glioblastoma by comparing RNA sequencing profiles of single glioblastoma cells and stem cells with profiles in its microenvironment. The research showed that mesenchymal, classical, and proneural subtype do in fact carry a distinct intrinsic transcriptional profile, while the same could not be said for the neuronal subtype which represents non-tumor cells in a tumor microenvironment [22]. Both research groups looked at the most common somatic mutations and copy-number alterations that were found in The Cancer Genome Atlas project (TCGA) [23]. TCGA research network highlights many altered genes, most of them could be characterized into one of 3 signaling pathways: receptor tyrosine kinase (RTK), retinoblastoma (RB), and p53. RTK signaling pathway was altered in 88% of sampled glioblastomas due to one or more mutations in *EGFR*, *NF1*, and *PTEN* genes. The RB signaling pathway was affected in 77% of glioblastomas sampled, including alteration of *CDKN2 A/B*, *CDK4*, *CDK6*, *CCND2*, and *RB1* genes. The last pathway, p53, was affected in 87% of observed glioblastoma samples through mutations in *CDKN2A*, *MDM2*, *MDM4* and *TP53* genes [24].

However, research shows that there is more to glioblastoma initiation, progression, and therapy resistance than that defined solely by its genetic component. The overall malignancy of the tumor is also influenced by the dynamic communication to its immediate and distant environment. Therefore, in this review, we describe the roles that extracellular vesicles (EVs) play in inflammation, and for “sending a message” from glioblastoma cells to the neighboring cells in the tumor microenvironment. These components seem to be crucial for ensuring cell survival and further tumor progression.

## 2. Extracellular Vesicles in Glioblastoma

EVs is a generalized term for particles that are naturally released from the cell in numerous biological fluids and extracellular space. They are not able to replicate, are encapsulated with a lipid bilayer [25] and loaded with cargo composed of proteins, lipids, carbohydrates, and nucleic acids [26]. The EVs should be described by their physical characteristics such as size and density, biochemical composition, biogenesis, or description of cell of origin. The terms such as “exosome” and “microvesicle” are to be avoided, as they lack unambiguous definition [25]. However, in most of the research papers these terms are used, and it is, therefore, useful to provide the basic differences in physical characteristics and biogenesis between distinct EVs. Exosomes are EVs that are of 30–100 nm in size, and originate from the endosomal network. Microvesicles are of 50–2000 nm in size, and arise through direct outward budding and fission of the plasma membrane. Apoptotic bodies are of 50–5000 nm in size, and arise from the cells in apoptosis [27].

Exosomes and microvesicles carry nucleic acids including miRNAs, mRNAs, other noncoding RNA and DNA, while apoptotic bodies can be characterized by the presence of organelles, chromatin, and glycosylated proteins inside the vesicles. The RNAs in the EVs are shorter than the ones in the cell, typically less than 200 nucleotides, but also longer transcripts can be present. The RNAs enriched in the EVs retain their functionality in the recipient cells. Although the RNAs have been the predominantly researched cargo in the EVs, the DNA functional role still must be determined. Protein exosome markers, important for exosome formations, are tetraspanins CD9, CD63, and CD81 while Alix, TSG101, HSC70, and HSP90β are part of the complexes named endosomal sorting complex required for exosome formation and transport. Although these proteins are thought to be exosomal markers, they can be found or present in other cells and processes [27,28,29,30]. So far, the markers for microvesicles have not been so well established, but they can carry the tetraspanins CD9, CD63, and CD81, and ARF6 and VCAMP3 have been proposed [27,29]. Recognized markers for apoptotic bodies are annexin V, thrombospondin, and C3b. The apoptotic bodies are usually cleared locally by the macrophages, which is mediated by the specific changes in the membrane composition and specific recognition receptor proteins on the macrophages. The most pronounced change in the membrane of the apoptotic bodies is the translocation of phosphatidylserine to the outer lipid bilayer, which bind to annexin V, further recognized by phacogytes. Membrane molecules can be also oxidized, leading to creation of binding sites for thrombospondin or the complement protein C3b, which are again recognized by phagocytes [27,30]. In the apoptotic bodies also HSP60 from mitochondrial origin or GRP78 from endoplasmic reticulum have been found to be enriched [29]. Beside interrogation of ‘‘classical’’ markers for EVs, also markers for oncogenic EVs exist. As such EGFR, EGFRvIII, GFAP, hTERT, IDH1, and IDH1R132 have been recognized. In recent papers on glioblastoma-derived EVs their cargo and corresponding biological role is elaborated in more detail [29,31].

The EVs are secreted by most cells, normal and cancerous, [27,32] and are perceived as cell-to-cell communication sources on long and short distances, transferring biological information through, e.g., oncogenes or nucleic acids [33]. These effect the recipient cells and play important roles in tumorigenesis, proliferation of cells, progression, metastasis as well as drug resistance [30]. The contents of the EVs reflects the state of the secreting cell, and oncogenic processes are suggested to increase the rate of the EVs release [27]. The EVs are being intensely investigated also in association with immune system, inflammation, their ability to modulate lymphocytes and macrophages.

In the brain the EVs are released by both neurons and different types of glial cells [26]. Over the period of 48 h approximately 10,000 EVs are released from a single glioma cell [34]. The EVs are also able to cross the blood–brain barrier in either direction, and are detected peripherally [26]. For instance, Zhuang et al. showed that exosomes loaded with drugs were able to cross the blood–brain barrier and were successfully taken up by microglia cells. The drug encapsulated in the EVs induced the apoptosis of the microglial cells and significantly delayed brain tumor growth [35].

The biomolecules packed in the EVs provide simultaneous delivery of multiple different messages from the cell of origin to target cell [26]. The tumor-derived EVs have been reported to be involved also in one of the critical events in glioblastoma progression —the angiogenesis. For instance, Skog et al. showed that microvesicles released by glioblastoma tumor cells are loaded with mRNA, miRNA, and angiogenic proteins. They demonstrated that mRNAs from the vesicles that were taken up by brain microvascular endothelial cells were translated, while the angiogenic proteins evoked tubule formation by endothelial cells in the tumor environment. Proteins angiogenin, IL-6, IL-8, TIMP-1, VEGF and TIMP-2 had higher concentrations in EVs than in the glioblastoma cells, and the first three were already previously associated with angiogenesis and increased malignancy in glioma, implying that microvesicles angiogenic effect could be partially promoted through angiogenic proteins [36]. Further on, the miR-148a packed in exosomes has been associated with cell proliferation and metastasis, through targeting CADM1 to activate STAT3 pathway [37]. For more detailed list of exosome involvement in glioblastoma development and progression please refer to Wu et al. [34].

Moreover, it has been shown that tumor-derived EVs also have high immunosuppressive activity. They can change the immune microenvironment, impacting chemo- and immune checkpoint blockers therapy. As such, cargos LGALS9 [38] and CD73 were identified [39]. In the central nervous system microglial cells EVs were determined to carry pro-inflammatory cytokine IL-1β and therefore to propagate inflammation [40].

## 3. Inflammation

### 3.1. Inflammation and Cancer

Inflammation is a complex process through which the immune system can respond to a stimulus. Injuries and infections are most commonly associated with inflammation. Additionally, inflammation can also stem from other type of internal disturbances, for example chronic inflammation associated with many chronic disease states, including cancer. Depending on the source of inflammation, distinctive cell types are activated. Compared to infections and injuries, where leukocytes and plasma protein are the key players, tissue macrophages seem to be more important in inflammation caused by internal disturbances, where inflammation process is not induced by bacteria or other external factors, but by endogenous signals of affected cells that are long-lasting and persistent. These signals can range from oxidized lipoproteins, advanced glycation end products, reactive oxygen species activity, extracellular matrix breakdown and others [41,42].

Inflammation appears to be a strong component of brain cancer as well, including glioblastoma. Tumor growth is accompanied by an increased hypoxia and aberrant vascular proliferation. On top, multiple types of immune cells can infiltrate the tumor microenvironment. These cells communicate using cytokines. With injuries or infection this would be an appropriate physiological response, but in cancer this combination of active inflammatory cells and mediators promotes tumor progression. Cytokine signaling can stimulate macrophages and other immune system cells, which in response produce reactive oxygen species such as superoxide and hydrogen peroxide, leading to an increased intracellular oxidative stress. When further interacting with molecules inside the cell, reacting oxygen species can interact with DNA, causing DNA oxidation damage (such as formation of 8-oxo-2′-deoxyguanosine, 5-chlorocytosine and others), leading to genetic mutations. Additionally, such changes can lead to disruption of epigenetic pattern as DNA damage can inhibit the binding of methyl-binding domain proteins [42,43,44].

Meta-analyses are an important tool for systematic assessment of research findings. One of the obstacles they face is the need for a moderate number of studies available for meta-analysis. Still, they provide an additional assessment of significance and a better estimate of the effect of findings [45]. In one such meta-analysis performed by Yang et al. the authors examined in detail the prognostic value of various systemic inflammation markers in glioblastoma patients. Altogether 18 studies were included in the meta-analysis, of which eight studies (1225 patients) examined the ratio of neutrophil to lymphocyte. Elevated values of neutrophils and increased ratio of neutrophil to lymphocyte have already been associated with poor prognosis of various cancer types. The meta-analysis confirmed this for glioblastoma, as in all glioblastoma patients a higher neutrophil to lymphocyte ratio predicted a worse prognosis. It appears that cancer cells and neutrophils work in a two-way mechanism; neutrophil-secreted cytokines help the tumor to progress while tumor cells secret chemotactic factor that can increase neutrophil count. In addition, Yang et al. reported the association of elevated platelet count with worse overall survival of glioblastoma patients [46].

### 3.2. Inflammation and Extracellular Vesicles

EV present a dynamic and efficient mean of communication between cells, but also between cancerous cells and their surroundings. To date, a smaller number of studies have examined the role of EV in glioblastoma development and progression. EV are a novel and evolving field of research, so most studies have been done in the past few years. These studies are summarized in Table 1. EV can carry different types of cargo, with studies mostly focusing on protein and RNA (predominately miRNA). The role of EV is also being investigated in glioblastoma treatment such as bevacizumab and TMZ. In a study by Simon et al. glioblastoma cells were treated with bevacizumab, a monoclonal humanized antibody that can neutralize VEGF-A, produced by tumor cells. When treating glioblastoma cells with bevacizumab, there was an observed difference in production and signaling of EVs. Tumor cells appeared to secrete bevacizumab via EVs as a way of trying to avoid the anti-angiogenic effect of therapy. When production of EVs was inhibited, the viability of glioblastoma cells was affected [47]. A recent study by Panzarini et al. investigated glioblastoma-secreted EVs in the presence or absence of TMZ, the most commonly used chemotherapeutic. When comparing TMZ-treated and non-treated cells, difference was observed in EVs amount, size, profile, and molecular signature of cargo. Expression of EVs markers decreased in TMZ-treated cells [48].

## 4. Glioblastoma Interactions with Its Microenvironment

Glioblastoma is a complex network of different tumor and stromal cells that interact with each other. Glioblastoma is in its cellular milieu or so-called tumor microenvironment [58]. Tumor interactions with its microenvironment are crucial for the progression of glioblastoma. Glioblastoma microenvironment is highly immunosuppressive [59] and consists of different cells, such as astrocytes, neurons, cancer stem cells, endothelial, immune and tumor-associated stromal cells that can promote tumor progression and therefore aid in therapy resistance [60,61,62]. Glioblastoma cells attract microglia, monocytes, and macrophages by secreting chemokines, growth factors, cytokines, matrix proteins, but also with tunneling nanotubes (gap junctions), and microtubes [52]. Besides these, the complex glioblastoma microenvironment is also characterized by acidosis and hypoxia [61]. A significant component of glioblastoma is the perivascular niche where glioblastoma stem cells can be found. The perivascular niche is composed of different cell types (pericytes, astrocytes, microglia, fibroblasts, and endothelial cells) that promote tumor progression. Moreover, to promote its survival glioblastoma also interacts with the extracellular matrix. The extracellular matrix is a dynamic compartment where components are deposited, degraded, or remodeled, which is critical for biological mechanisms.

Glioblastoma stem cells can recruit immunosuppressive cells to the tumor microenvironment which then support stem cell phenotypes, chemoresistance, evasion of immune surveillance and invasion [61]. Such cells are tumor-associated macrophages (TAMs), myeloid-derived suppressor cells (MDSCs), T-regulatory (Treg) cells, and natural killer (NK) cells. Under the influence of cancer cells and tumor microenvironment, macrophages become tumor-associated macrophages (TAMs) that promote glioblastoma growth [59]. Normally, M1 macrophages are capable of phagocytosis, cytotoxicity, antigen presentation and secretion of inflammatory cytokines. In glioblastoma however, M1 convert to M2 macrophages that produce angiogenic factors, immunosuppressive molecules, EVs, as well as chemokines, cytokines and growth factors that favor angiogenesis and tumor progression [63,64]. The role of exosomes in the conversion of M1 into M2 macrophages is described in detail by Baig et al. [65]. M1 and M2 macrophages have different markers. In particular, IL-6, TNF-α, and IL-12 are specific for M1, while CD163, CD206, IL-10, and TGF-β are specific for M2 macrophages [66].

As mentioned above and illustrated in Figure 1, glioblastoma cells communicate with their immediate and distant environment through various metabolites, but also with the help of EVs [59]. EVs can modulate the immune microenvironment and other physiological processes, and can interact with the host immune system [65,67]. EVs secreted by glioblastoma cells contain different biological molecules [44]. Once they reach the recipient cell, they can be internalized and deliver their message [52]. For example, Gabrusiewicz et al. studied the mechanism by which exosomes induce transformation of macrophages to the M2 type [56]. The authors showed that exosomes secreted by glioblastoma cells reach monocytes, are internalized into their cytoplasm and cause reorganization of their cytoskeleton. In addition, they provided evidence that monocytes take up exosomes secreted by glioblastoma cells that release signal transducer and activator of transcription 3 (STAT3) which induces expression of PD-L1 and polarization of the macrophages to the M2 tumor-supportive phenotype. One of the mechanisms is by binding of PD-L1, which is found on tumor and antigen presenting cells, to programmed cell death protein 1 (PD1) which is found on activated T-cells, therefore blocking T-cell activation and inhibiting T-cell killing of tumor cells [68]. In the study by Ricklefs et al. the authors examined the mechanism of EV-mediated suppression of T-cell activation and showed that PD-L1 found on EVs can directly ligate PD1. Ricklefs et al. provided proof for a novel mechanism of glioblastoma evasion of the immune system. In a different study, van der Vos et al. studied how glioma cells manipulate microglia and macrophages from their environment by releasing extracellular RNA [52]. They observed that in the presence of EVs derived from glioma cells, microglia showed increased proliferation of 40% over 7 days. The authors also reported altered cytokine profile in microglia exposed to glioma-EVs.

In the study by Azambuja et al. the authors examined the effect of glioblastoma-derived exosomes on macrophages [69]. They were able to show in vitro that glioblastoma-derived exosomes can reprogram naïve pro-inflammatory and antitumoral (M1) macrophages and promote their transformation to M2 anti-inflammatory and protumoral macrophages, but can also convert M2 macrophages to strongly immunosuppressive TAMs. Conversion of naïve to M1 macrophages was confirmed with the expression of the phenotypic markers CD80, CD86, major histocompatibility complex class II (HLA-DR), and interferon (INF)-γ. Conversion into M2 macrophages was proven with increased expression of arginase-1, IL-10, and CD206, and decreased expression of the M1 markers described previously. In addition, M2-like macrophages showed increased cell migration. Therefore, the authors were able to prove that glioblastoma-derived exosomes induce pro-tumor phenotype in different classes of macrophages. This resulted in release of so-called “secondary” exosomes by TAMs and consequently promotion of tumor growth. In another publication from the same authors, Azambuja et al. showed that chemoresistant glioma cells modulate the polarization of M2 macrophages so that it favors tumor recurrence and progression [70]. The authors used in vitro and in vivo models to evaluate the crosstalk between glioma cells and macrophages. Macrophage polarization was validated with IL-10 release, CD206 expression and arginase activity. They reported that chemoresistant glioma cells induce stronger immunosuppressive macrophage polarization than chemosensitive glioma cells that at the end resulted in tumor proliferation. As published by Qian et al. glioblastoma-derived exosomes affect microphage polarization to M2 type, and promote tumor progression in vitro and in vivo [66]. Hypoxia is one of the factors that determine the contents of exosomes, and can induce polarization of TAMs. With microRNA sequencing the authors identified miR1246 as enriched in hypoxic glioma-derived exosomes and able to induce M2 polarization by targeting TERF2IP. This results in the activation of STAT3 and inhibition of NF-κβ signaling pathway, leading to promotion of glioblastoma proliferation. Similarly, Svensson et al. showed the hypoxic glioblastoma cells secrete microvesicles loaded with tissue factor (TF)/VIIa [71]. This triggers up-regulation of protease activated receptor 2 (PAR-2) and increases levels of pro-angiogenic growth factor heparin-binding EGF-like growth factor (HB-EGF). Moreover, Kucharzewska et al. showed that exosomes secreted by glioblastoma cells mediate intracellular communication in hypoxic conditions [72]. They suggest that exosomal molecular signature CAV1, IL8, PDGFs, and MMPs, can be a noninvasive, biomarker profile that reflects glioblastoma hypoxic signaling, and can be used to assess its oxygenation status.

Using a bioinformatics approach, Jia et al. identified a cohort of 44 genes that are associated with glioblastoma microenvironment and can predict poor patient outcome [58]. The authors used data from TCGA project, and validated the results using data from the Chinese Glioma Genome Atlas (CGGA). For the identified differentially expressed genes they performed functional enrichment clustering that showed strong association with immune response. With the STRING tool, the authors obtained protein-protein interaction networks and identified three main modules: IL6, TIMP1, and TLR2, reported to promote angiogenesis, proliferation, migration, and invasiveness. In the TLR2 module several proteins and their corresponding genes, in particular TLR2, CCL2, CCL5, IGSF6, and CD14, critical for immune response were clustered.

Although glioblastoma cells are crucial for tumor initiation and progression, their interactions with the microenvironment are very important for keeping the tumor alive.

## 5. Therapy

Knowledge of EVs in glioblastoma is of particular interest as EVs are involved in different major cancer stages such as initiation and progression [34], and could be therefore used as biomarkers or even diagnostic biomarkers detectable in peripheral tissues, such as blood or cerebrospinal fluid. For instance, the tumor-specific EGFRvIII has been detected in EVs from serum in 7 (25%) out of 25 glioblastoma patients, adding important diagnostic information as it is associated with specific subtypes of glioma, while on the other hand also representing an additional support in diagnostic decisions [36].

On the other hand, EVs could be used as the innovative strategy for glioblastoma treatment. Specifically, the chemotherapeutic drug treatments achieve insufficient drug concentrations due to blood–brain barrier restrictions. Therefore, an alternative approach is gravely needed. Compared to synthetic polymers, virus-based vectors or lipids EVs have higher delivery efficiency and better biocompatibility, making them promising nanocarriers [73]. Due to the small size and presence of the surface molecules, they have high tissue affinity and natural target capacity, and consequently less of the undesired off-target effects [30]. They represent a potential for cell-free therapy, and are rather easily manipulated. They can be isolated directly from patient’s body fluid or from patient’s cultured cells, followed by desired modification and transferred back to the patient [74].

The exosomes can be used as vehicles to deliver antitumor miRNAs, proteins, or other molecules; however, first the identification of the EVs cargo involved in the cancer initiation, progression, chemoresistance, and immunosuppression must be identified.

Several different mechanisms through which exosomes, and their cargo proteins and nucleic acids, promote or influence tumor growth through chemoresistance, have been proposed: (i) influencing the recipient cells, (ii) effective drug use limitation, (iii) chemoresistance phenotype transfer and (iv) reshaping of tumor microenvironment [32]. One of the most important issues in glioblastoma treatment is tumor chemoresistance, and the exosomes have been shown to influence the target cells to promote chemoresistance. Some noncoding RNA molecules transmitted through exosomes have been identified as molecules that promote glioblastoma chemoresistance to commonly used chemotherapeutic agent TMZ. As such, long noncoding RNA (lncRNA) HOX transcript antisense intergenic RNA (HOTAIR), already previously associated with cancer proliferation [75], was found to be significantly up-regulated in TMZ-resistant glioblastoma cells, inducing TMZ resistance and modulated TMZ resistance through miR-519a-3p/RRM1 axis. HOTAIR down-regulation inhibited proliferation, migration, invasion, and epithelial/mesenchymal transition in TMZ-resistant glioblastoma cells [76]. Similarly, also lncRNA SBF2-AS1A transmitted through exosomes causes TMZ resistance [77]. Among exosome transmitted miRNAs that have been associated with TMZ resistance the miR-1238 has been identified. It was found to confer chemoresistance in the tumor microenvironment [78]. Munoz et al. found two miRNAs, miR-93 and -193, to be present only in TMZ-resistant glioblastoma cell lines and primary spheres. The two miRNAs probably target Cyclin D1, which is a major regulator of cell-cycle progression [79]. On the other hand, miR-151a transferred through exosomes sensitized TMZ-resistant glioblastoma cells through XRCC4-mediated DNA repair pathway. Due to its mechanism of action the miR-151a represents a promising therapeutic target [80].

To overcome the chemoresistance caused by exosomes two strategies exist. Either the exosome biogenesis has to be inhibited or the exosomes are used as the delivery vehicles [32]. For example, Jia et al. prepared exosomes loaded with superparamagnetic iron oxide nanoparticles (SPIONs) and curcumin. They further conjugated the exosome membrane with neuropilin-1-targeted peptide to achieve glioma-targeting exosomes. The prepared exosomes could cross the blood–brain, showed good results for targeted imaging, while SPIONs and curcumin provided a potent synergistic antitumor effect [81]. Munoz et al. first reported that increase in miR-9 caused TMZ-resistant glioblastoma cells. The miR-9 is involved in the expression of the drug efflux transporter, P-glycoprotein. In their further experiments they showed that transfer of mesenchymal stem cells microvesicles loaded with anti-miR-9 to the resistant glioblastoma cells reversed the expression of the multidrug transporter and sensitized the glioblastoma TMZ-resistant cells [82].

Till today number of studies reporting on EVs loaded with either anti-cancer drug or biomolecule which could be used as glioblastoma therapy remains scarce. Further research of suitable target molecules as well as more physical issues concerning exosomes loading capacity, proper dosage, and targeted delivery must be addressed. Currently there are several ongoing clinical trials on glioma and miRNAs, while it is still too early for clinical studies of EVs applications. More details are in Table 2.

## 6. Conclusions

Communication between glioblastoma cells and microenvironment stimulates growth and proliferation, and protects the tumor from immunosurveillance and chemotherapy. Together with chemokines and cytokines, EVs are part of this communication. The role of EVs in communication between glioblastoma cells and their environment is currently being explored. There are indications that through EVs glioma cells can influence their neighboring calls and change their phenotype. Still, biologically active concentrations of EVs that actually have a function are yet to be determined. Once this is elucidated, EVs and their bioactive cargo can be modified to serve a different purpose such as disease treatment or delivery of bioactive molecules and drugs. Yet, in vivo validation of the in vitro findings must be performed.

Although the EVs are becoming an important field of molecular biology and genetics of cancer, the methodology for EVs preparation and manipulation remain rather inconclusive and therefore not yet suitable for translation into clinical practice. More research in this filed is needed to standardize the procedures that would provide protocols of sufficient reproducibility for wider use. However, further dissection of the EVs biology is important to use the EV as biomarkers, as it is now clear that EVs possess the ability to manipulate the tumor environment, allowing the tumor growth and making the tumors more invasive. The accumulated knowledge will gradually contribute also to the development of EVs as therapeutic approach, as an alternative to current treatments is imperative.

## Figures and Tables

**Figure 1 biomedicines-10-00151-f001:**
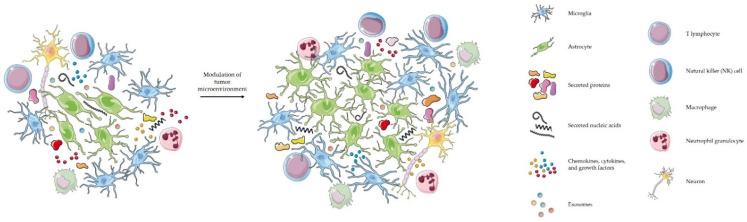
Dynamic communication between glioblastoma and its environment. Schematic illustration of secretion by different molecules by glioblastoma cells, and their influence on modulating the tumor microenvironment. Excretion of various biologicals promotes glioblastoma progression. The illustration is for graphical presentation only and does not represent actual sizes or size ratios among particles. The illustration was created using Servier Medical Art (SMART) (https://smart.servier.com/, accessed on 19 November 2021 and 04 January 2022). Servier Medical Art by Servier is licensed under a Creative Commons Attribution 3.0 unsupported license.

**Table 1 biomedicines-10-00151-t001:** Studies examining the role of extracellular vesicles and inflammation in glioblastoma patients.

Type of Extracellular Vesicle	Cargo Type	Tissue/Cell Type	Main Outcome	Author
EVs	RNA	Plasma and serum samples of patients	Difference in EVs cargo from plasma and EVs cargo from serum (269 and 636 differentially expressed genes in plasma and serum). Changes in plasma EVs associated with inflammation, changes in serum EVs associated with ubiquitinylation and cytokine signaling.	Roy et al., 2021 [49]
Exosomes and microvesicles	miRNA	GL261 mouse glioma cell line	Glioma-derived vesicles can be transported to microglia which was demonstrated using miR-21. In microglia there was a down-regulation of miR-21 target genes, resulting in increased microglia proliferation.	Abels et al., 2019 [50]
Exosomes	miRNA	U87 and P3 human glioma cell lines U87 and P3 mouse glioma cell lines	Increased formation of exosomes under hypoxic conditions compared to normal. Hypoxia-derived exosomes induced more myeloid-derived suppressor cells. Exosomal miR-10a and miR-21 induced the expansion and activation of myeloid-derived suppressor cells via RORA and PTEN pathway.	Guo et al., 2018 [51]
EVs (majority 100–200 nm)	miRNAs	Primary human glioblastoma cells mouse microglia	Glioblastoma-isolated EVs were taken up by microglia, resulting in increased proliferation. Cytokine profile trend toward immunosuppression. Most abundant miRNAs in vesicles were miR-451 and miR-21. Both miRNA target c-Myc mRNA, which decreased in microglia that uptook EVs.	van der Vos et al., 2016 [52]
Exosomes	Protein	BATF2-overexpressing glioma cell lines GBM patient plasma	BATF2 is involved in inflammatory antitumor response. It inhibits the recruitment of myeloid-derived suppressor cells.BATF2 positive exosomes as a potential biomarker (distinction between stage III-IV vs. stage I-II vs. healthy subjects).	Zhang et al., 2021 [53]
Exosomes	Protein	U87MG and T98G Glioma stem cells	Addition of a selective COX-2 inhibitor leads to a change in function of secreted exosomes from glioma stem cells (decreased adherent cell migration of U87MG and T98G).	Palumbo et al., 2020 [54]
Microvesicles	Protein	Human and mouse tissue	Myeloid-derived suppressor cells can promote regulatory B-cell function via microvesicles. Microvesicles contained PD-L1, resulting in the ability of regulatory B-cell to suppress the CD8þ T-cell activation.	Lee-Chang et al., 2019 [51]
EVs	Protein	Patient-derived glioblastoma stem cells	Glioblastoma-derived EVs were associated with changes in astrocyte proteome. In-silico prediction of MYC, NFE2L2, FN1, and TGFβ1 activation, and p53 inhibition, leading to a tumor-favoring phenotype of astrocytes.	Hallal et al., 2018 [55]
Exosomes	Protein	Glioblastoma-derived stem cells	Secreted exosomes are taken up by monocytes, which results in phenotypic change to immunosuppressive M2 macrophages.	Gabrusiewicz et al., 2018 [56]
Exosomes	Protein	U373 glioma cells	Increased levels of CRYAB when stimulated with IL-1b and TNF. Changes in composition of the secreted exosomal proteome when stimulated with cytokines.	Kore et al., 2014 [57]

Abbreviations: BATF2—basic leucine zipper ATF-like transcription factor 2; COX-2—cyclooxygenase-2; CRYAB—heat shock protein CryAB; EVs—extracellular vesicles; FN1—Fibronectin 1; IL-1b—interleukin-1 beta; MYC—MYC Proto-Oncogene; NFE2L2—NFE2 Like BZIP Transcription Factor 2; p53—tumor protein P53; PD-L1—programmed death-ligand 1; PTEN—phosphatase and tensin homolog; RORA—RAR-related orphan receptor alpha; TGFβ1—Transforming Growth Factor-β1; TNF—tumor necrosis factor-alpha.

**Table 2 biomedicines-10-00151-t002:** Ongoing clinical trials examining the miRNAs of glioma patients.

Clinical Trial Status	Study Title	Conditions or Disease/Biological Sample	ClinicalTrials.gov Identifier
Recruiting	Blood Biomarker Signature in Glioma	Glioma/Serum	NCT03698201
Recruiting	Evaluating the Expression Levels of MicroRNA-10b in Patients with Gliomas	Astrocytoma Oligodendroglioma Oligoastrocytoma Anaplastic Astrocytoma Anaplastic Oligodendroglioma Anaplastic Oligoastrocytoma Glioblastoma Brain Tumors Brain Cancer/Blood Tumor tissue Cerebrospinal fluid	NCT01849952
Not yet recruiting	LIQUID BIOPSY IN Low-grade Glioma Patients (GLIOLIPSY)	Glioma/Blood samples	NCT05133154
Recruiting	Multicenter Safety Trial Assessing an Innovative Tumor Molecular and Cellular Print Medical Device in Glioma (ProTool)	Oligodendroglioma Astrocytoma/Brain tissue	NCT02077543
Recruiting	Research on Precise Immune Prevention and Treatment of Glioma Based on Multi-omics Sequencing Data	Glioma/Peripheral blood Urine Feces Glioma tissue Brain tissue Meninges Cerebrospinal fluid	NCT04792437
Recruiting	Determination of Immune Phenotype in Glioblastoma Patients	Glioblastoma Multiforme/Peripheral Blood Mononuclear Cell Plasma Tumor tissue	NCT02751138
Recruiting	A Phase II/III Study of High-dose, Intermittent Sunitinib in Patients with Recurrent Glioblastoma Multiforme (STELLAR)	Glioblastoma Multiforme/Blood	NCT03025893
Recruiting	Molecular Genetic, Host-derived, and Clinical Determinants of Long-term Survival in Glioblastoma	Glioblastoma/Blood	NCT03770468

## Data Availability

Not applicable.

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
