# Peer review of "Dynamic Intercell Communication between Glioblastoma and Microenvironment through Extracellular Vesicles"

_biomedicines, 2022, doi:10.3390/biomedicines10010151_

Round 1
Reviewer 1 Report
The manuscript is well written, it sums up various aspects about extracellular communication between glioblastoma cells and neighboring cells in the TME, as well as some systemic effects mediated by EVs.
The modulation of M1 macrophages to M2-TAM is a very interesting mechanism commonly shared with other malignancies (i.e. PCs), and it is mentioned the role of the PD-L1 expression in that specific context regarding to macrophages. Although there are evidences of PD-L1 expression on glioblastoma-derived EVs (which are enriched upon stimulation of glioblastoma cells with pro-inflammatory cytokines), with the potential to directly bind to PD1 on the surface of T-cells. This mechanism has been shown to be capable of T-cell modulation of T-cells tumor-suppressing activity according to Ricklef et al (2018 manuscript). It would be nice to add those informations in the current review for overall completion even without pushing too far into the lymphocytes topic.
Reviewer 2 Report
The manuscript reviews the genetic composition of glioblastoma and its microenvironment, in which vesicles (EVs) mediate interaction between tumor cells and stroma cells as well as immune suppression. Moreover, EV roles in inflammation and its therapeutic application are discussed.
1.EV type (exosomes, micro vesicles, etc) and content classification and their roles (protein, microRNA, mRNA, etc) need to be elaborated in more details.
- In addition to glioblastoma cells, the other cell types in the microenvironment contribute to EVs production that has been shown in some papers. This is good to be included for discussion.
- Are there more recent research using EVs for targeting delivery and some clinical trials ongoing? This information is helpful to support the potential of EVs application to therapy.
- The tumor cells and immune cells, etc need to be added to the Figure 1 schema.
- The number of glioma cells needs to be given for producing 10,000 EVs over the period of 48 hours on line 33 of page 3.
- Some grammar error and typo: line 12 on page 7 – a space should be given between the sentences; line 17 on page 8 – “that” should be “than’; line 23 on page 23 – inconsistency with past tense in the previous sections.
